# Prognostic Value of Copy Number Alteration Burden in Early-Stage Breast Cancer and the Construction of an 11-Gene Copy Number Alteration Model

**DOI:** 10.3390/cancers14174145

**Published:** 2022-08-27

**Authors:** Dingyuan Wang, Songlin Gao, Haili Qian, Peng Yuan, Bailin Zhang

**Affiliations:** 1Department of Breast Surgery, National Cancer Center/National Clinical Research Center for Cancer/Cancer Hospital, Chinese Academy of Medical Sciences and Peking Union Medical College, Beijing 100021, China; 2Department of VIP Medical Services, National Cancer Center/National Clinical Research Center for Cancer/Cancer Hospital, Chinese Academy of Medical Sciences and Peking Union Medical College, Beijing 100021, China; 3State Key Laboratory of Molecular Oncology, National Cancer Center/National Clinical Research Center for Cancer/Cancer Hospital, Chinese Academy of Medical Sciences and Peking Union Medical College, Beijing 100021, China

**Keywords:** breast cancer, copy number alteration burden, prognosis

## Abstract

**Simple Summary:**

Breast cancer is a malignancy that poses a significant threat to women’s health. The enormous disease burden has forced a wide range of researchers to develop more accurate prognostic models. Copy number alterations, which are amplifications or deletions of DNA fragments, often predict a poor prognosis. Instead, copy number alteration burden, i.e., the level of CNA, may have a good predictive value for disease prognosis. In this study, we developed a prognostic model for early breast cancer based on CNAB and simplified it. It performed excellently in two external validation sets.

**Abstract:**

The increasing burden of breast cancer has prompted a wide range of researchers to search for new prognostic markers. Considering that tumor mutation burden (TMB) is low and copy number alteration burden (CNAB) is high in breast cancer, we built a CNAB-based model using a public database and validated it with a Chinese population. We collected formalin-fixed, paraffin-embedded (FFPE) tissue samples from 31 breast cancer patients who were treated between 2010 and 2014 at the National Cancer Center (CICAMS). METABRIC and TCGA data were downloaded via cBioPortal. In total, 2295 patients with early-stage breast cancer were enrolled in the study, including 1427 in the METABRIC cohort, 837 in the TCGA cohort, and 31 in the CICAMS cohort. Based on the ROC curve, we consider 2.2 CNA/MBp as the threshold for the CNAB-high and CNAB-low groupings. In both the TCGA cohort and the CICAMS cohort, CNAB-high had a worse prognosis than CNAB-low. We further simplified this model by establishing a prognostic nomogram for early breast cancer patients by 11 core genes, and this nomogram was highly effective in both the TCGA cohort and the CICAMS cohort. We hope that this model will subsequently help clinicians with prognostic assessments.

## 1. Introduction

Breast cancer is the most prevalent malignancy in women. According to the National Cancer Center of China, the estimated number of new breast cancer cases in China is as high as 304,000 [1]. The increasing burden of breast cancer has prompted a wide range of researchers to search for new prognostic markers [2]. With the promotion of next-generation sequencing technology, an increasing number of multigene models are being established and used in clinical practice due to their accuracy compared to traditional clinical models [3,4].

Somatic cell copy number alterations (CNAs) are one of the hallmarks specific to malignancy, and represent the amplification or deletion of a DNA fragments [5]. The CNA of a gene implies genomic instability and often predicts a poor prognosis [6,7,8,9]. It is generally accepted that CNA increases with increasing cancer stage and is higher in patients with advanced breast cancer than in patients with early-stage breast cancer [10]. Copy number alterations of individual genes are often the result of altered chromosomal segments. Takayuki pointed out that loss at 6q, 13q, and 16q, as well as gain at 1q, 6p, 8q, 9p, 11q, 16p, 17q, 19q, and 20q in patients with breast cancer can predict high chromosomal instability, tumor immune escape, and strong tumor aggressiveness [11,12]. Copy number alteration burden (CNAB), as a total level of CNA, will better indicate chromosome stability as well as tumor prognosis. Tumor mutational burden (TMB), another indicator of genomic stability, is often used in prognostic studies of tumors. Breast cancer is one of the tumor types with low mutation frequency, resulting in low TMB with low variability. The mutation frequencies of PI3KCA and TP53, the two most commonly mutated genes in breast cancer, were 34.5% and 34.3%, respectively, and nonsynonymous TMB was only 1.20 Muts/Mb [13]. A study by Liu’s team included eight cancer cohorts, including breast cancer (*n* = 1234), for survival analysis. This study found no significant difference in overall survival between patients with high TMB and low TMB (*p* = 0.351) [14]. However, there was a significant difference in overall survival between breast cancer patients in the high-CNAB and low-CNAB cohorts (*p* = 0.004). In addition, CNAB is also a significant predictor of survival for tumor patients with many other cancer types [15,16,17]. Considering the high frequency of CNA and low frequency of mutation in breast cancer, we believe that the utilization of CNA as a prognostic marker is a very promising topic.

Although transcriptome-based early breast cancer prognostic models continue to be used in the clinic with well-recognized effectiveness, with the improvement of liquid biopsy technology, ctDNA-based prognostic models may be more widely used in the future due to their advantageous features (e.g., they are less invasive and provide reproducible measurements). Compared with transcriptome-based prognostic models, the establishment of CNAB-based prognostic models would be more conducive to the development of future ctDNA prognostic models, thereby allowing clinicians to make prognostic assessments more conveniently.

Admittedly, some studies have revealed that CNAB is a prognostic factor for early-stage breast cancer [2,6,10,18]. However, prognostic evaluation based on whole-exome CNAB would obviously place a great cost burden on patient and be very inconvenient for laboratory physicians. Therefore, we attempted to optimize an all-exon CNAB model. A traditional method of screening for core genes is to find the genes most associated with prognosis by statistical methods. However, this can lead to potential false-positive results. Considering that CNAs are more likely to affect biological pathways only if they alter the transcription of genes, we performed a secondary screen among prognosis-related CNAs, eliminating genes that are not strongly linked to mRNA expression. We finally obtained an 11-gene model to accurately assist clinicians and facilitate treatment decisions.

## 2. Materials and Methods

### 2.1. Ethics

The study methodologies conformed to the standards set by the Declaration of Helsinki and were approved by the Ethics Committee of the National Cancer Center/National Clinical Research Center for Cancer/Cancer Hospital, Chinese Academy of Medical Sciences and Peking Union Medical College (NCC-2021C-369). All patients signed an informed consent form in writing.

### 2.2. Study Design

This retrospective study collected formalin-fixed, paraffin-embedded (FFPE) tissue samples from 31 breast cancer patients who were treated between 2010 and 2014 at the Cancer Hospital, Chinese Academy of Medical Sciences (CICAMS). The patients included all had early invasive breast cancer. Resection specimens rather than core biopsies were used for DNA extraction. The METABRIC cohort and TCGA cohort are two large breast cancer cohorts, and we downloaded clinical information for both cohorts via the cBioPortal website (https://www.cbioportal.org, accessed on 1 July 2022).

For the CICAMS cohort, we retrieved the patients’ age, pathological details, and treatment information from their medical records. The main variables in this analysis were (a) the patients’ demographic characteristics (age, etc.) and (b) their clinical information (grade, ER status, PR status, HER2 status, surgery, radiotherapy, and chemotherapy). The stage of breast cancer was categorized based on the American Joint Committee on Cancer, Seventh edition [19]. The grade was categorized as grade I to III based on the WHO [20]. Age was categorized as younger than 60 years or 60 years and older. Those with missing or unclear records were categorized as unknown. For the METABRIC and TCGA cohorts, we retrieved the following clinical variables: age, grade, stage, molecular subtype, and treatment strategies.

### 2.3. Genomic Information Acquisition

Details of the extraction of DNA and next-generation sequencing are available in the Appendix A. CNA, TMB, and mRNA data of the METABRIC cohort and TCGA cohort were obtained from https://www.cbioportal.org. We investigated the transcriptional data by fragments per kilobase of transcript per million mapped reads (FPKM) values. We used these transcriptomic data for subsequent differentially expressed gene analysis and gene ontology analysis. For the assessment of CNA, we used the GISTIC 2.0 criterion [21]. This criterion uses a fixed algorithm to transform the amplification or deletion status of each gene into an integer between −2 and 2.

### 2.4. CNAB11 Modeling Method

We used the METABRIC training cohort to obtain prognosis-related CNAs by differential CNA analysis with 5-year recurrence/metastasis as the endpoint event. Then, we further screened reliable prognosis-related CNAs with a strong association with mRNA using the Kolmogorov–Smirnov test. Finally, 11 genes (CBWD1, CDC6, CWC25, HS3ST3A1, IFNA2, KDM4C, KRT27, MLLT6, NBR1, NBR2, and ZDHHC21) were included in the CNAB11 model. The model combined these 11 gene CNAs (GISTIC 2.0) to obtain the CNAB11 score. The cutoff for the CNAB11 cluster was then obtained from the receiver operating characteristic curve.

### 2.5. Statistical Analysis

The differences in clinicopathological characteristics and treatment strategies were compared through the chi-square test or Fisher’s exact test for categorical variables and the Wilcoxon rank sum test for ordered variables. Among the groups, prognostic differences were estimated with the log-rank test for categorical variables. Cox regression was used to calculate hazard ratios (HRs) and their 95% confidence intervals (CIs). Relapse-free survival (RFS) was defined as the time from radical resection for breast cancer to the earliest time of recurrence or death from any cause. Overall survival (OS) was defined as the time from the date of diagnosis to the date of death due to any cause or to the last follow-up [22]. Based on the gene expression data of the TCGA cohort, we used the “DESeq2” package in R to analyze the differentially expressed genes (DEGs) between the two subgroups. The screening criteria for DEGs were *p* < 0.05 and absolute log2FC > 2. We used the Kolmogorov–Smirnov test to examine the consistency of the distribution of CNA and mRNA. All analyses were performed in R 4.0.1 (https://www.r-project.org/, accessed on 1 July 2022). The R package “RCircos” was used to generate the circle graphs. GraphPad Prism 6 (https://www.graphpad.com/scientific-software/prism/, accessed on 1 July 2022) was adopted to plot the survival curves. Two-sided tests were used for all analyses. A *p* value less than 0.05 was considered statistically significant.

## 3. Results

### 3.1. Patient Characteristics

A total of 2295 patients with early-stage breast cancer were enrolled in the study, including 1427 in the METABRIC cohort, 837 in the TCGA cohort, and 31 in the CICAMS cohort. The mean age of the patients in the METABRIC cohort was 60.68 years (standard deviation: 12.97 years). A total of 496 (34.76%) patients in the METABRIC cohort were stage I, 816 (57.18%) were stage II, 115 (8.06%) were stage III, 942 (66.01%) received radiotherapy, 1034 (72.46%) received drug therapy, and 818 (57.32%) received mastectomy. The mean age of patients in the TCGA cohort was 58.65 years (standard deviation: 13.18 years). A total of 147 (17.56%) patients in the TCGA cohort were stage I, 494 (59.02%) were stage II, and 196 (23.42%) were stage III. A total of 439 (52.45%) patients in the TCGA cohort received radiotherapy, 65 (7.77%) received drug therapy, and 382 (45.64%) received mastectomy. The mean age of patients in the CICAMS cohort was 53.44 years (standard deviation: 10.02 years). Of these, 16 (51.61%) were stage II, 15 (48.39%) were stage III, 13 (41.94%) received radiotherapy, 27 (87.1%) received medication, and all received mastectomy (Table 1). 

In addition, the median CNABs of patients in the METABRIC, TCGA, and CICAMS databases were 2.1, 3.8, and 1.5 CNA/Mbp, respectively (Figure 1a). The genes with more than 50% CNA in patients from METABRIC (Figure 1b), TCGA (Figure 1c), and CICAMS (Figure 1d) are presented using circle plots. The percentage of CNA per chromosome for these three cohorts is also presented in bar graphs (Figure 1e–g).

### 3.2. Differences in Clinical Characteristics between Different CNAB Subgroups

Patients in METABRIC were divided into a training cohort (*n* = 714) and a test cohort (*n* = 713). Receiver operating curve (ROC) analysis was performed in the METABRIC training cohort with 5-year OS, 10-year OS, 5-year RFS, and 10-year RFS as the endpoints. The 5-year RFS had the highest AUC value (Appendix A, AUC = 0.611) when the CNAB threshold was 2.2 CNA/Mbp. Therefore, we divided the cohorts into a CNAB-high group and a CNAB-low group according to whether the CNAB was higher than 2.2 CNA/Mbp.

The differences in clinical characteristics between the CNAB-high group and CNAB-low group of the METABRIC cohort are shown in Appendix A. In the test cohort, there were 341 patients with hormone receptor (HoR)+ human epidermal growth factor receptor 2 (HER2)-, 34 with HER2+, and 43 with triple-negative breast cancer (TNBC) in the CNAB-low group, and 169 with HoR+HER2−, 58 with HER2+, and 68 with TNBC in the CNAB-high group. The differences between the two groups were statistically significant (*p* < 0.001). The CNAB-low group had a lower grade than the CNAB-high group (*p* < 0.001).

The differences between the CNAB-high and CNAB-low groups in the TCGA cohort are shown in Appendix A. There were 213 (84.86%) patients in the CNAB-low group with a molecular subtype of HoR+HER2−, higher than in the CNAB-high group (*p* < 0.001). The CNAB-low group had a lower stage than the CNAB-high group (*p* = 0.006).

### 3.3. Survival Analysis between CNAB Groups

In the METABRIC test cohort, the CNAB-high group had a worse prognosis than the CNAB-low group, with both shorter RFS and shorter OS (Figure 2a,b). After adjusting for age, subtype, grade, stage, and treatment strategies, the multivariate Cox regression model showed a 33% (RFS) and 25% (OS) higher risk for the CNAB-high group compared to the CNAB-low group (Appendix A). Similarly, in the TCGA cohort, the CNAB-high group showed a worse prognosis than the CNAB-low group in terms of both RFS and OS (Figure 2c,d). Adjusted by age, stage, and treatment strategy, the CNAB-high group had shorter RFS (HR = 1.62, 95% CI: 1.08–2.46, *p* = 0.021) and shorter OS (HR = 1.94, 95% CI: 1.15–3.28, *p* = 0.013) than the CNAB-low group (Appendix A). In the CICAMS cohort, the CNAB-high group had a worse prognosis for RFS than the CNAB-low group (HR = 6.28, 95% CI: 1.08–36.84, *p* = 0.042) (Figure 2e).

Furthermore, we used different cutoff values to dichotomize CNAB, and the cutoff was taken as 2.2 or 2.6 CNA/Mbp as a valid predictor of prognosis in both the METABRIC test cohort and TCGA cohort. When the threshold is below 2.2 or above 2.6 CNA/Mbp, there will no longer be a statistically significant difference in RFS and OS in the CNAB-high group compared to the CNAB-low group (Appendix A). We also performed survival analyses for different subgroups (Appendix A). Some subgroups no longer had statistically significant survival differences due to the reduction in the number of events after subgrouping.

### 3.4. Combined Survival Analysis of TMB and CNAB

We performed a combined survival analysis of TMB and CNAB in the TCGA cohort. TMB was higher in the CNAB-high group according to the Wilcoxon test (*p* < 0.001) (Appendix A). The TCGA cohort was divided into four groups according to the quartiles of TMB. Survival analysis showed no prognostic value for RFS or OS (Appendix A). Then, the cohort was divided into high and low groups based on the median TMB. Combined survival analysis with CNAB showed that the TMB-high/CNAB-low group had the best prognosis for OS, which was statistically significant (*p* = 0.039) (Appendix A).

### 3.5. Construction and Validation of the CNAB11 Model

In the METABRIC training cohort, chi-squared tests were performed for whole-exon CNAs between the 5-year relapse group and the 5-year nonrelapse group, with a total of 449 genes selected for corrected *p* values < 0.05 (Appendix A). Meanwhile, we conducted a concordance test of CNA and mRNA for each patient in the METABRIC training cohort, with a total of 726 genes screened for the Kolmogorov–Smirnov test *p* < 0.05 (Appendix A). We took the intersection of these two gene sets to obtain 11 genes (CBWD1 (9p24.3), CDC6 (17q21.2), CWC25 (17q12), HS3ST3A1 (17p12), IFNA2 (9p21.3), KDM4C (9p24.1), KRT27 (17q21.2), MLLT6 (17q12), NBR1 (17q21.31), NBR2 (17q21.31), and ZDHHC21 (9p22.3)). We summed the number of copy number changes for these 11 genes to obtain the CNAB11 score. In the METABRIC training cohort, we used ROC to conclude that the cutoff should be taken as 6.

We divided the METABRIC, TCGA, and CICAMS cohorts into high and low groups based on the CNAB11 score. Survival analysis showed that in the METABRIC test cohort, the CNAB11-high group had a significantly worse prognosis than the CNAB11-low group (RFS: HR = 1.35, 95% CI: 1.09–1.66, *p* = 0.005; OS: HR = 1.25, 95% CI: 1.01–1.56, *p* = 0.044). Similarly, in the TCGA cohort, the prognosis was worse in the CNAB11-high group than in the CNAB11-low group (RFS: HR = 1.55, 95% CI: 1.08–2.23, *p* = 0.017; OS: HR = 2.59, 95% CI: 1.46–4.59, *p* = 0.001). In the CICAMS cohort, the CNAB11-high group had a shorter RFS than the CNAB11-low group (HR = 5.94, 95% CI: 1.08–32.72, *p* = 0.017) (Figure 3).

Then, we built the nomogram in the METABRIC training cohort based on a multivariate Cox regression model (Figure 4a). In the TCGA cohort and the CICAMS cohort, we calculated the score for each patient individually based on the nomogram. The 5-year RFS was then predicted for both cohorts. ROC showed that the nomogram was a good predictor of 5-year RFS in patients in both the TCGA cohort (AUC = 0.72) and the CICAMS cohort (AUC = 0.83) (Figure 4b).

### 3.6. Differences in CNA and Expression Profiles of Different CNAB11 Groups

We analyzed the CNA differences between the CNAB11 high and CNAB11 low populations using Fisher’s test. For the TCGA cohort, the differential CNA between the CNAB11-high and CNAB11-low populations was distributed on almost all chromosomes, but was most significant on chromosomes 6, 9, 17, and 20 (Figure 5a). For the CICAMS cohort, differential CNA was scattered on most chromosomes, but most densely on chromosome 14 (Figure 5b).

We analyzed the expression profiles of the CNAB11-high group and CNAB11-low group in the TCGA cohort to screen for differentially expressed genes (Appendix A). The heatmap (Figure 6a) and volcano map (Appendix A) are shown. We further analyzed the differentially expressed genes by Gene Ontology (GO), and the results showed that they were enriched in 33 GO terms, such as “positive regulation of establishment of protein localization to telomere” (Figure 6b).

## 4. Discussion

With the above results, it is evident that CNAB has better prognostic prediction than TMB for early breast cancer patients. Furthermore, our simplified CNAB11 model is similar to the CNAB model in that both have good predictive effects. The METABRIC cohort is dominated by loss, while CICAMS is dominated by gain. However, CNAB is a metric that homogenizes METABRIC and CICAMS well. The CNAB and CNAB11 models built by the METABRIC cohort can be used for the CICAMS cohort, even though they have different CNA types. Therefore, we believe that the CNAB-based model has generalizability across different racial cohorts.

Several studies have confirmed the close association between somatic copy number alterations in some genes and the prognosis of patients [15,16,17]. In this paper, we also found prognosis-associated CNAs in breast cancer. Notably, the genes in CNAB11 were mainly distributed at 17q and 9p. This is consistent with the findings of Budczies and Ueno [11,12]. Therefore, in the future, if histological specimens are used for prognostic evaluation, fluorescence in situ hybridization can be considered for the corresponding regions. Since ctDNA tends to be fragmented, we believe that it may be more accurate to use the corresponding 11 probes for detection if this model is to be used for liquid biopsies in the future.

Somatic cell copy number alteration burden has also been demonstrated as a new prognostic marker in many cancer types [2,6,8,10,14,18,23,24,25]. Few investigators have focused on the predictive role of copy number variant burden on the prognosis of patients with early-stage breast cancer. In fact, CNAB has more predictive potential for prognosis than TMB due to the higher rate of copy number alterations than mutations in breast cancer. Therefore, we believe that CNAB-based prognostic models should be investigated and explored by more researchers.

For early-stage breast cancer, there are several widely accepted prognostic models based on gene expression profiles, such as Oncotype DX. The TAILORx study showed that Oncotype DX can accurately predict survival in ER (+), HER2 (−), and axillary lymph node-negative breast cancer, thus guiding the choice for chemotherapy [26]. The expression levels of genes are continuous variables that are more suitable for accurate modeling than mutations or copy number alterations. With the improvement of DNA sequencing technology in recent years, blood tissue-based ctDNA testing has become increasingly mature [27,28]. Wang et al. concluded that the DNA sequencing results of tumor tissue specimens and blood specimens are similar [29]. Therefore, prognostic models based on TMB or CNAB are expected to be tested noninvasively with ctDNA in the future. Surgical resection specimens are difficult to obtain for both early-stage breast cancer patients receiving neoadjuvant therapy and advanced breast cancer patients, at which point, prognosis-specific CNA by blood ctDNA is an excellent option. Thus, we suggest that CNAB-based prognostic models are very promising in all cancer types. In addition, DNA sequencing-based models can be used in conjunction with transcriptome-based models. Both models complement each other and may play an increasingly important role in the prognosis of early breast cancer patients in the future.

However, the cost of whole-exome sequencing is becoming increasingly low. Nevertheless, for some patients, it is not appropriate to perform whole-exome sequencing for prognostic assessment. Therefore, we simplified the CNAB model and selected 11 promising genes. Kong and Mahadevappa demonstrated experimentally that CDC6 plays an important role in the progression of breast cancer and can help predict the prognosis of breast cancer patients [30,31]. Several studies also found that HS3ST3A1 was a novel tumor regulator and a good predictor of survival in both lung cancer patients and breast cancer patients [32,33]. In breast cancer, glioma, and colorectal cancer, KDM4C is involved in biological processes such as tumorigenesis and metastasis [34,35,36]. MLLT6 has been confirmed to play an important role in immune maintenance [37]. NBR1 and NBR2 are closely related to the regulation of BRCA1 and have important roles in the genesis and progression of many tumors [38]. Few studies have shown the prognostic value of CBWD1, CWC25, IFNA2, KRT27, or ZDHHC21 in breast cancer, and subsequent studies are expected. There are two potential limitations to this article. First, the number of patients in CICAMS was relatively small, which may have resulted in potential selection bias. Second, the 11CNAB model we constructed was not validated by further animal experiments, and we will follow up with related research.

## 5. Conclusions

For early-stage breast cancer, CNAB is a better prognostic factor than TMB and has shown great results in the European-dominated METABRIC cohort, the American-dominated TCGA cohort, and the CICAMS cohort. We subsequently constructed an 11-gene CNA-based prognostic model. This model was shown to have promising prognostic indications in both the TCGA cohort and the CICAMS cohort.

## Figures and Tables

**Figure 1 cancers-14-04145-f001:**
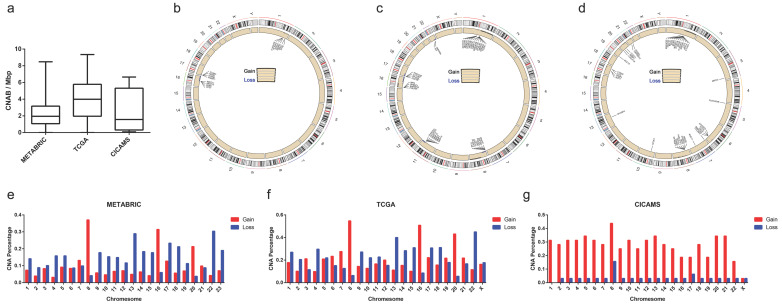
Copy number alteration of the study population. (**a**) Copy number alteration burden of three breast cancer cohorts; (**b**–**d**) the genes with more than 50% copy number alteration in patients from METABRIC, TCGA, and CICAMS; (**e**–**g**) the percentage of CNA per chromosome from METABRIC, TCGA, and CICAMS.

**Figure 2 cancers-14-04145-f002:**
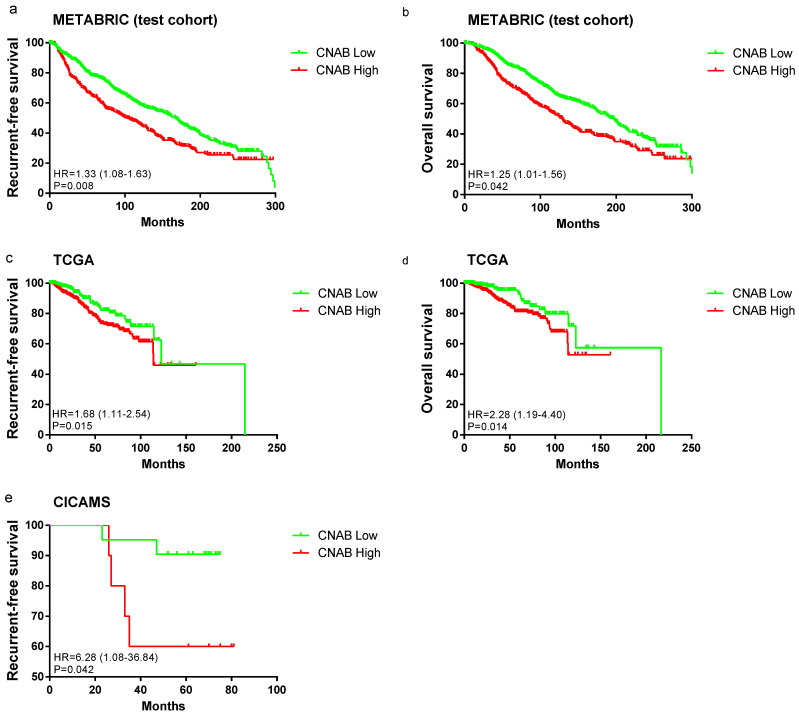
Survival analysis on relapse-free survival and overall survival of patients in the (**a**,**b**) METABRIC test cohort, (**c**,**d**) TCGA, and (**e**) CICAMS between the CNAB-high group and CNAB-low group.

**Figure 3 cancers-14-04145-f003:**
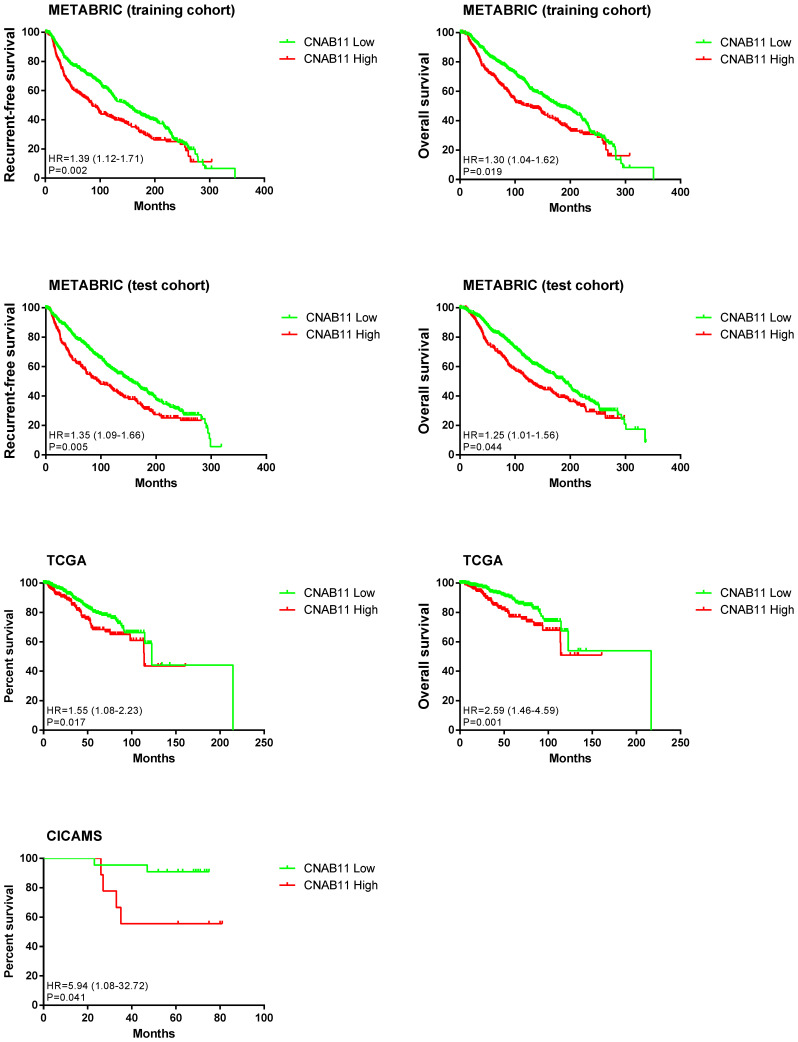
Survival analysis on relapse-free survival and overall survival of patients in METABRIC, TCGA, and CICAMS between the CNAB11-high group and CNAB11-low group.

**Figure 4 cancers-14-04145-f004:**
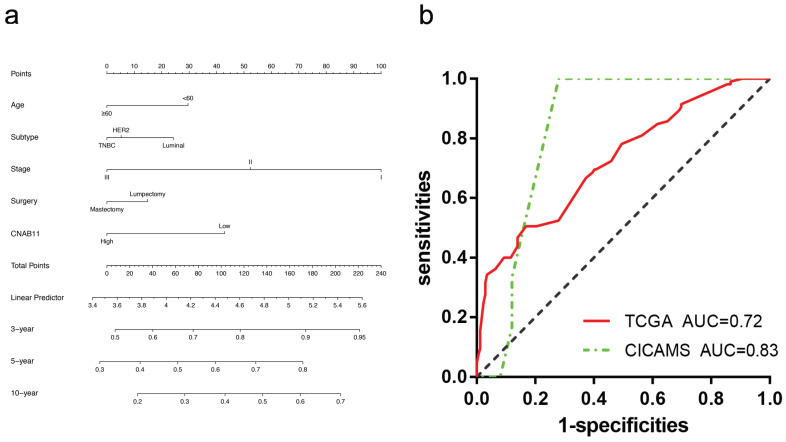
(**a**) Nomogram of early-stage breast cancer based on CNAB11 score and clinical characteristics. (**b**) Receiver operating curve of 5-year relapse-free survival in TCGA and CICAMS.

**Figure 5 cancers-14-04145-f005:**
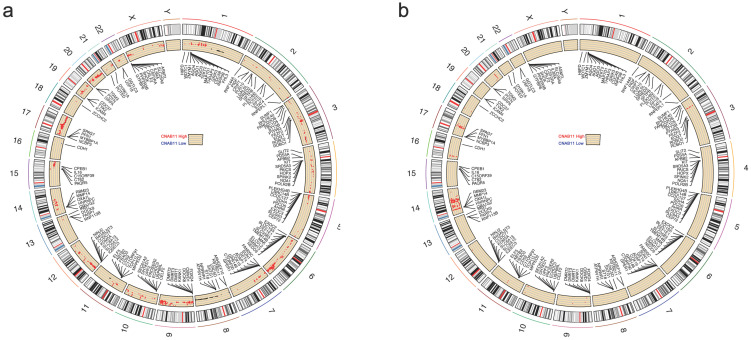
The differential CNA between the CNAB11-high and CNAB11-low populations in (**a**) the TCGA cohort and (**b**) the CICAMS cohort.

**Figure 6 cancers-14-04145-f006:**
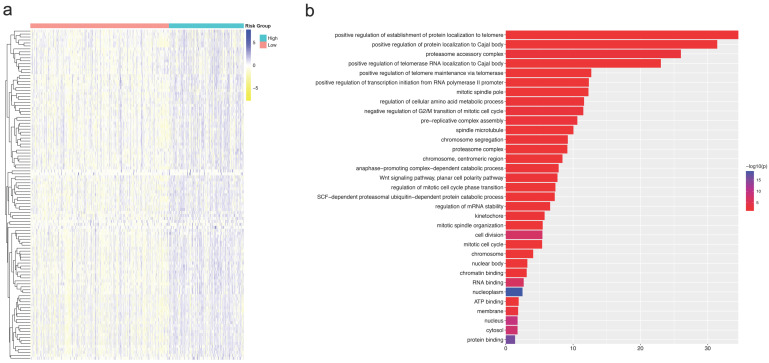
(**a**) Heatmap of the expression profile difference between the CNAB11-high group and the CNAB11-low group. (**b**) Gene ontology analysis between the CNAB11-high group and the CNAB11-low group.

**Table 1 cancers-14-04145-t001:** Clinical characteristics and treatment strategies.

	METABRIC		TCGA		CICAMS	
	Cases	%	Cases	%	Cases	%
Overall	1427		837		31	
Age						
<60	647	45.34%	445	53.17%	22	70.97%
≥60	780	54.66%	392	46.83%	9	29.03%
Subtype						
HoR ^†^ +HER2 ^‡^ −	1025	71.83%	527	62.96%	15	48.39%
HER2+	177	12.4%	174	20.79%	14	45.16%
TNBC ^§^	225	15.77%	136	16.25%	2	6.45%
Grade						
I	116	8.13%			0	0%
II	549	38.47%			17	54.84%
III	715	50.11%			14	45.16%
Stage						
I	496	34.76%	147	17.56%	0	0%
II	816	57.18%	494	59.02%	16	51.61%
III	115	8.06%	196	23.42%	15	48.39%
Radiotherapy						
Yes	942	66.01%	439	52.45%	13	41.94%
No	485	33.99%	398	47.55%	18	58.06%
Drug therapy						
Yes	1034	72.46%	65	7.77%	27	87.1%
No	393	27.54%	772	92.23%	4	12.9%
Chemotherapy						
Yes	308	21.58%			24	77.42%
No	1119	78.42%			7	22.58%
Hormone Therapy						
Yes	875	61.32%			14	45.16%
No	552	38.68%			17	54.84%
Surgery						
Mastectomy	818	57.32%	382	45.64%	31	100%
Lumpectomy	609	42.68%	194	23.18%	0	0%
Unknown						

^†^ HoR, hormone receptor; ^‡^ HER2, human epidermal growth factor receptor 2; ^§^ TNBC, triple-negative breast cancer.

## Data Availability

All data generated or analyzed during this study are available from the corresponding author upon reasonable request.

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
