# Peer review of "Prognostic Value of Copy Number Alteration Burden in Early-Stage Breast Cancer and the Construction of an 11-Gene Copy Number Alteration Model"

_cancers, 2022, doi:10.3390/cancers14174145_

Round 1

Reviewer 1 Report

The purpose of this manuscript is clear and it gives interest to readers of "Cancers", however, the number of Chinese cohort is relatively small. Therefore, the validation using CICAMS data is necessary in the future. Anyway, the result in this manuscript is promising.

11genes which have been used in this model should be mentioned in Materials and Methods.

The reason of selecting 11genes should be mentioned in Introduction

In Introduction, full spelling of CNAB is better to be described. It means  copy number alteration burden (CNAB)

Reviewer 2 Report

The authors have developed a new prognostic model for breast cancer. Their model is based more on copy number alteration burden and less on transcriptional changes or tumor mutation burden. The authors have to expand on their results and make interpretations from their data. This is lacking is many figures. 

1. The english language needs to be checked by a native english speaker. Grammatical errors in the text need to be fixed. 

2. The introduction needs to be expanded to include more details about CNV in breast tumors. What chromosomes are changed? Is their difference between CNV in early vs late tumors. What is the tumor mutation burden seen in breast tumors? Which genes are commonly mutated (BRCA)? How does particular chromosome change give advantage to tumor for its survival? These points are relevant to this study and need to be discussed in the introduction. These also will help authors provide rationale for their study. 

3. CNABs are main focus of the paper. Supplementary fig 1 which talks about CNAB should be a main figure. Also, authors don't discuss specific chromosome numbers and changes within them. They look at bulk changes but individual changes needs to be addressed. Can authors get that from their data?

4. How does mastectomy or lumpectomy change CNABs within breast tumors? is there a pattern (change in specific chromosomes) amongst patients who have undergone surgery vs those who haven't? 

Round 2

Reviewer 2 Report

The authors have addressed my comments and I don't have any further suggestions